# Optimization of Feedback FET with Asymmetric Source Drain Doping Profile

**DOI:** 10.3390/mi13040508

**Published:** 2022-03-25

**Authors:** Inyoung Lee, Hyojin Park, Quan The Nguyen, Garam Kim, Seongjae Cho, Ilhwan Cho

**Affiliations:** 1Department of Electronic Engineering, Myongji University, Yongin-si 17058, Gyeonggi-do, Korea; 98chapssal@naver.com (I.L.); oct95@naver.com (H.P.); quinn523@mju.ac.kr (Q.T.N.); garamkim@mju.ac.kr (G.K.); 2Department of Electronic Engineering, Gachon University, Seongnam-si 13120, Gyeonggi-do, Korea

**Keywords:** feedback field-effect transistor (FBFET), device optimization, on–off current ratio, subthreshold swing, TCAD

## Abstract

A feedback field-effect transistor (FBFET) is a novel device that uses a positive feedback mechanism. FBFET has a high on-/off ratio and is expected to realize ideal switching characteristics through steep changes from off-state to on-state. In this paper, we propose and optimize FBFET devices with asymmetric source/drain doping concentrations. Additionally, we discuss the changes in electrical characteristics across various channel length and channel thickness conditions and compare them with those of FBFET with a symmetric source/drain. This shows that FBFET with an asymmetric source/drain has a higher on-/off ratio than FBFET with a symmetric source/drain.

## 1. Introduction

Metal-oxide-semiconductor field-effect transistors (MOSFETs) have been scaled down over forty years to achieve a high density, low power consumption, and high on-current [1,2,3,4]. In order to improve the performance and integration of MOSFETs, research on reduction has been in progress for a long time [3,4,5,6]. Various problems, such as an increase in leakage current and drain-induced barrier lowering, have been raised in the reduction in MOSFETs [2,3,4,5,6,7]. In order to overcome such MOSFET problems, various studies have attempted to change the device structure or operation mechanism [8,9,10,11]. The devices that have been proposed to overcome the limitations of MOSFETs include a tunneling field-effect transistor (TFET) [12,13,14], impact-ionization MOSFET (i-MOSFET) [12,13,14,15,16,17], nano electron mechanical field-effect transistor (NEMFET), and ferroelectric negative-capacitance field-effect transistor (NCFET) [18,19,20].

Among novel devices, the feedback field-effect transistor (FBFET) creates a potential well inside a channel and operates through a feedback phenomenon due to electron and hole movement. MOSFET cannot have subthreshold swings (S) below 60 mV/dec, whereas FBFET can have S below 60 mV/dec and a relatively high on-current level [21]. To improve FBFET’s performance characteristics, such as the on/off ratio and S value, FBFET devices with various structures have been studied. A nanowire (NW) FBFET [22], which contains a nanowire channel, can obtain a large on/off ratio and small S value compared with conventional FBFET. The use of a two-stack silicon on insulator (SOI) FBFET and three-stack SOI FBFET [23] has been proposed; these can obtain a high on-current by increasing the effective channel width by stacking the gate all around (GAA) channel. Optimization through changing the channel material has also been studied. When using Si${}_{1-x}$Ge${}_{x}$ as an FBFET device’s channel material, it is possible to operate at a lower drain voltage than the Si channel. Thus, a reduction in power consumption can be achieved [24].

If a nanowire structure is adopted or a non-silicon material is used, a high level of performance improvement can be obtained, but it is difficult to avoid increasing the process complexity and experiencing related problems. In this study, we describe research results that can improve the performance of the double gate FBFET, which is easier to manufacture than the NW FBFET, by optimizing the structure parameters of the device. In addition to the influence of conventional parameters on the device, guidelines for the performance improvement of FBFET devices are presented through an asymmetric source–drain structure.

## 2. Device Structure and Methods

A schematic diagram of the FBFET is shown in Figure 1a, and the mesh was set up for device simulation as described in Figure 1b. Mesh was set up tightly at 1 nm × 1 nm at the silicon region, where a feedback phenomenon occurred. To reduce the computational complexity, a minimum mesh size of 1 nm × 5 nm was used in other areas, such as the oxide, poly silicon gate region. The proposed device had a p-type and n-type poly silicon twin gate structure. The intrinsic (P-) channel formed a virtual junction through two different gates which had different dopants and work functions. This work was carried out with a two-dimensional simulation structure and the overall electrical characteristics and their variations were investigated using a technology computer-aided design (TCAD) simulator (Silvaco Atlas, version 5.20.2 R). To demonstrate the electrical characteristics of the device, several models were applied within the process of simulation. The models and parameters included Shockley–Read–Hall (SRH) recombination, a surface SRH model for the respective interface, Auger recombination resembling for mobility (affected by concentration and temperature), generation and recombination mechanisms, and carrier lifetime estimation. Simulations also included inter band tunneling and quantum insight models to compute the quantum tunneling. The impact ionization mechanism of FBFETs was found to be negligible in previous studies, so it was not considered in this study [11,22,25,26].

The important device parameters used in the simulation are listed in Table 1. Figure 1c shows the transfer characteristics of the device with the default device parameters. It can be seen that steep switching occurred at the place where the off-current region changed to the on-current region, and a sufficient on–off current ratio of 10${}^{7}$ could be obtained. Additionally, as shown in Figure 1d, an extremely short time of within 25 μs was required to change the off-state to on-state. This work represents an attempt to create structures utilizing commonly used materials to achieve a high performance. Thus, the device was made using silicon (Si) or silicon-product materials; specifically, for the source, drain, and channel we considered the use of Si with different doping types and concentrations. The gate dielectric material used was silicon dioxide (SiO${}_{2}$), which was thought to be more reliable and compatible. However, it is expected that the optimization data obtained in this study can be applied even if the gate oxide is changed to a high-k dielectric.

Figure 2 describes the approximate fabrication process used for the proposed FBFET device. It would have been hard to create the device using the conventional MOSFET fabrication process, since the device had two types of gate doped with n-type and p-type structures, respectively. Thus, it was necessary to form the n-type and p-type poly silicon gates separately. First, a fin-type silicon channel had to be prepared, as shown in Figure 2a. Subsequently, an iterative etching and deposition process was conducted two times, as shown in Figure 2b–e. A more intensive fabrication process was required for this compared with conventional MOSFET, but it was possible to achieve this with technology that already existed and the process was not expected to be complex.

## 3. Results and Discussion

### 3.1. Simulation Conditions

Since the drain voltage (V${}_{DS}$) value according to the channel length has been found to have a significant effect on the on–off current of FBFET [27], it is essential to determine the appropriate V${}_{DS}$ value before the simulation for optimization. Figure 3a shows the transfer characteristics of an FBFET with a channel length of 130 nm and a channel thickness of 22 nm when applied with various V${}_{DS}$ from 0.6 to 1 V. Similar to previous research results, V${}_{DS}$ had a significant effect on the electrical characteristics, including the on–off current, in our device. Figure 3b shows the S and on–off current ratio obtained with V${}_{DS}$ variation. The S values were less than 60 mV/dec, the limitation of S in MOSFETs, for a V${}_{DS}$ higher than 0.7 V. Compared to the previously published FBFET research results, the overall S value was relatively large. In this study, we extracted the value of S by finding the average of the slope in the transition section instead of the maximum value of the slope at the transient point. The reason why we used this method is that the average S (S${}_{avg}$) can accurately represent the change obtained with an operation voltage reduction [28,29]. When V${}_{DS}$ is greater than 0.7 V, the FBFET provides adequate switching characteristics with a high on-current (10 nA/μm). From the S value and the on–off current ratio, which have opposite change characteristics according to V${}_{DS}$, we can obtain a clue about the optimized V${}_{DS}$. However, when considering the operation of the device, it is necessary to consider the threshold voltage change as well. In Figure 3c, the threshold voltage (V${}_{TH}$) fluctuation does not show a significant change below 0.9 V of V${}_{DS}$ but decreases rapidly above 0.9 V. It can be seen that a value smaller than 0.9 V should be used to keep V${}_{TH}$ stable when determining the drain voltage. In terms of the on–off current ratio and S, the most optimized outcomes shown in Figure 3b occurred at V${}_{DS}$ of 0.95 V and 1 V, respectively. However, lowering the bias condition also is an important aspect considered in this study; henceforth, a V${}_{DS}$ of 0.8 is used for the rest of this paper, giving us a significantly high on–off current ratio, a small S, and a V${}_{TH}$ in the stable range.

### 3.2. Doping Variation

#### 3.2.1. Source and Drain Doping Variation

In this chapter, the optimized V${}_{DS}$ obtained based on the results of the previous chapter is applied in the simulation. We performed this experiment by fixing the other parameters as shown in Table 1 while varying the doping concentration of the source and drain simultaneously at a V${}_{DS}$ of 0.8 V and a V${}_{FG}$ of 0.45 V. As shown in Figure 4a, the change in the current in the OFF region is negligible because the change in the source drain doping has a relatively small effect on the barrier height of the electrons and holes formed at the virtual junction. However, a current difference is observed in the ON region. When the on-current flows in the FBFET, a feedback operation occurs and the barrier of electrons and holes disappears. In this case, the difference in the relative energy band height between the channel and the source drain determines the on-current level. For this reason, an asymmetric doping concentration dependence between the on-current and off-current occurs. This is also confirmed by the result of Figure 4b. Since the S value is determined by the switching characteristics, it is less affected by the source/drain doping concentration variation and the threshold voltage is also less affected, as shown in Figure 4c.

#### 3.2.2. Source Doping Variation with Fixed Drain Doping

Figure 5a shows the transfer characteristics with regard to the influence of a source doping variation from 5 × 10${}^{18}$ to 1 × 10${}^{20}$ cm${}^{-3}$ with the drain region set at 1 × 10${}^{19}$ cm${}^{-3}$. Compared with the result shown in Figure 4, the effect of source doping on the on-current change was insignificant, as shown in Figure 5a. In Figure 5b, the change in S was similar to the previous one, and the change in the on-/off current ratio was relatively small. Therefore, we can expect that the change in source doping has a slight effect on the electron and hole barriers and a small effect on the energy band even in the ON region after feedback occurs. In particular, the small effect on the barrier that affects the switching characteristics is confirmed by a slight change in the threshold voltage, as shown in Figure 5c. Therefore, it can be concluded that the change in source doping does not significantly affect the change in electrical characteristics.

#### 3.2.3. Drain Doping Variation with Fixed Source Doping

Figure 6 shows the simulated I${}_{DS}$–V${}_{GS}$ for various drain doping concentrations at a source doping of 1 × 10${}^{19}$ cm${}^{-3}$, V${}_{DS}$ of 0.8 V, and V${}_{FG}$ of 0.45 V. When only the drain doping is varied, the trend of on–off current change is almost the same as that when the source and drain doping are changed at the same time in Figure 4. In Figure 6a–c, these trends are all consistent. In this scenario, when the drain doping is changed, the drain doping cannot cause a change in the barriers that affect the off region, but it has significant influence on the factors that determine the on-current. Compared with the results shown in Figure 4 and Figure 5, it is clear that the drain doping variation is the main reason for the change in electrical characteristics. This represents the first attempt to analyze the effects of source–drain doping separately and may be able to provide deeper insight into FBFET optimization. In the case of a general MOSFET, the source and drain doping are of the same type, so changing the asymmetric doping requires a lot of effort. However, since FBFETs have asymmetric doping types, asymmetric doping optimization can be achieved without much effort.

### 3.3. Dimensional Feature Variation

In order to verify the feasibility of the FBFET having an asymmetrically doped source and drain, a comparison of the device’s characteristics with those of symmetrically doped FBFET was carried out while changing various device parameters. The FBFETs used for the comparison had a source doping of 1 × 10${}^{19}$ cm${}^{-3}$, and, in the case of drain doping, the asymmetric FBFET had a doping of 5 × 10${}^{18}$ cm${}^{-3}$ and the symmetric FBFET had one of 1 × 10${}^{19}$ cm${}^{-3}$.

#### 3.3.1. Channel Length Variation

Figure 7a,b show the transfer characteristics of symmetric and asymmetric FBFETs when the channel length was changed from 80 nm to 170 nm. A change in the channel length has a very important effect on achieving the high integration of the device. In order to analyze the electrical characteristics of the FBFET device in detail, the on–off current ratio, S, and V${}_{TH}$ were determined. In the comparison result obtained for the on–off ratio shown in Figure 7c, it can be seen that the FBFET with an asymmetric source–drain structure had superior characteristics compared to that with a symmetric structure. This is consistent with the previous results shown in Figure 4, where the difference is evident for short channel elements. This finding is significant with regard to scaling down the FBFET. In both devices, the S value was improved when the channel length was shortened. Although there was some variation, in the case of S the asymmetric-structure FBFET did not show much difference from the symmetric-structure FBFET, as shown in Figure 7d. In the case of V${}_{TH}$, there was no difference in the characteristics of the two FBFETs according to their channel length, as shown in Figure 7e.

#### 3.3.2. Channel Thickness Variation

This comparison between the symmetric and the asymmetric structures used a silicon channel thickness varying from 42 nm to 18 nm. In Figure 8a (the symmetric) and Figure 8b (the asymmetric), it is clear that the transfer characteristics are similar in both of the designs. In both FBFET structures, the on–off current ratios decreased because the off-current increased rapidly when the channel thickness increased, as shown in Figure 8c. In this case, the asymmetric-structure FBFET always maintained a higher on–off current ratio compared to the symmetric-structure FBFET.

In the previous study, when S was obtained at a particular point, there was no significant change in S according to the channel thickness variation [23]. Similar to the previous results, a steep slope in the transition curve can be seen in Figure 4a,b. However, in this study where the average S was obtained, S decreased as the channel thickness increased, as shown in Figure 8d. The asymmetric-structure FBFET has similar S characteristics to those of the symmetric-structure FBFET. It can be seen that there was no significant difference between the two FBFETs even in terms of the V${}_{TH}$ change, as shown in Figure 8e, which had the same pattern as the channel length variation. As shown in Figure 7 and Figure 8, the asymmetric FBFET was superior to the symmetric FBFET in terms of the on–off current ratio when the device dimensions were changed.

## 4. Conclusions

In this paper, the device optimization of FBFET was investigated through a TCAD simulation. In the observation of the device characteristics according to source and drain doping changes, the FBFET with asymmetric doping had superior characteristics compared to those of the FBFET with a symmetric structure. The asymmetric FBFET showed an excellent on–off current ratio among its various electrical characteristics, and, in the case of V${}_{TH}$ and S, the asymmetric and symmetric FBFET had similar characteristics. The asymmetric FBFET had the same superiority when the channel length and thickness of the device were changed. Therefore, it can be concluded that this structure is applicable for scaled-down devices. Device technology that can optimize FBFET without applying a new material or process was obtained in this work. Finally, the advantages obtained from the use of the asymmetric source–drain structure are expected to be applicable to the FBFETs of other structures.

## Figures and Tables

**Figure 1 micromachines-13-00508-f001:**
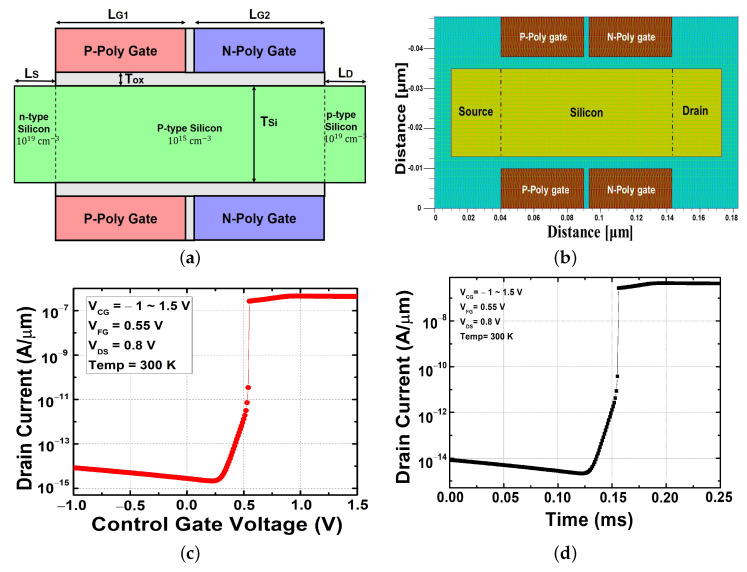
Proposed architecture: (**a**) scheme structure of dual-gate feedback field-effect transistor, (**b**) mesh set up for device simulation, (**c**) transfer characteristic, and (**d**) transient characteristic at the previous bias condition background.

**Figure 2 micromachines-13-00508-f002:**
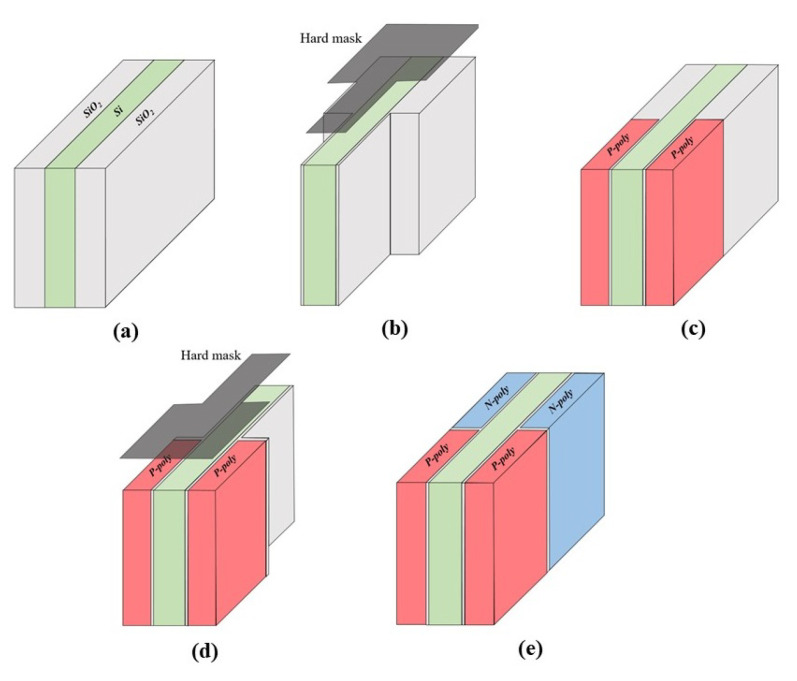
Fabrication process used for the proposed FBFET structure: (**a**) channel and gate oxide region formation, (**b**) double P-poly gate patterning using hard mask, (**c**) double P-poly gate deposition, (**d**) double N-poly gate patterning using hard mask, (**e**) double N-poly gate deposition.

**Figure 3 micromachines-13-00508-f003:**
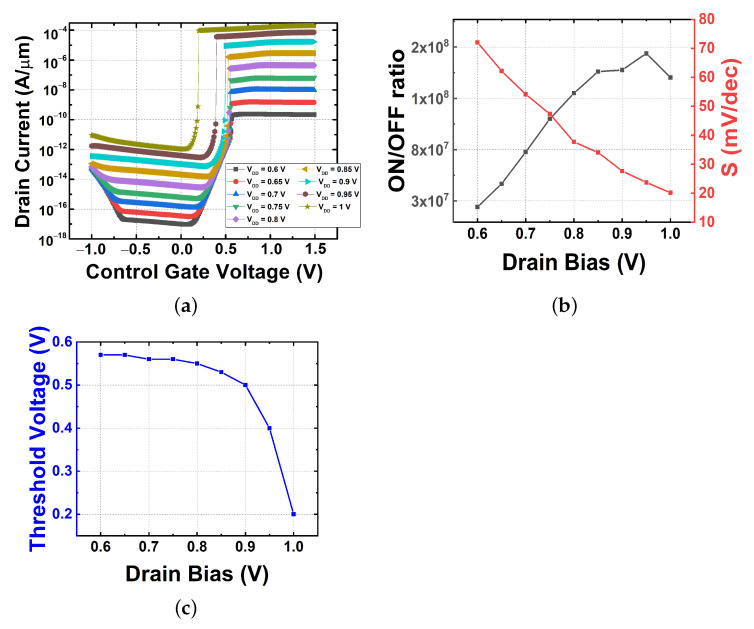
Electrical characteristics with drain voltage variations. (**a**) Transfer characteristics, (**b**) on–off current ratio and subthreshold swing, and (**c**) threshold voltage.

**Figure 4 micromachines-13-00508-f004:**
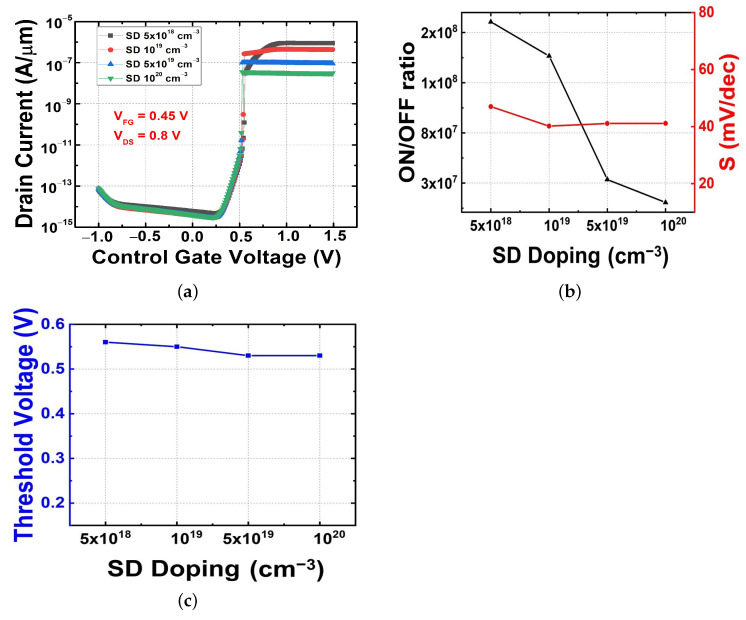
Electrical characteristics obtained with source drain doping variations. (**a**) Transfer characteristics, (**b**) on–off current ratio and subthreshold swing, and (**c**) threshold voltage.

**Figure 5 micromachines-13-00508-f005:**
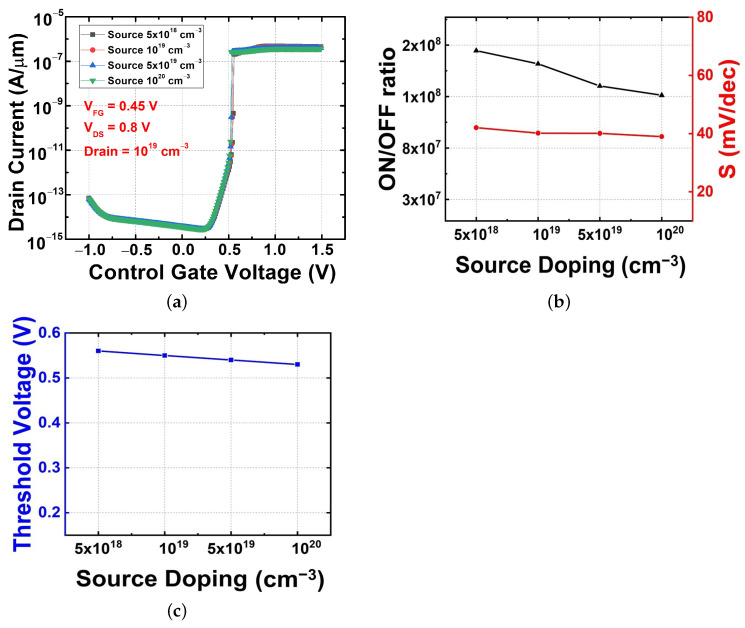
Electrical characteristics with source doping variations. (**a**) Transfer characteristics, (**b**) on–off current ratio and subthreshold swing, and (**c**) threshold voltage.

**Figure 6 micromachines-13-00508-f006:**
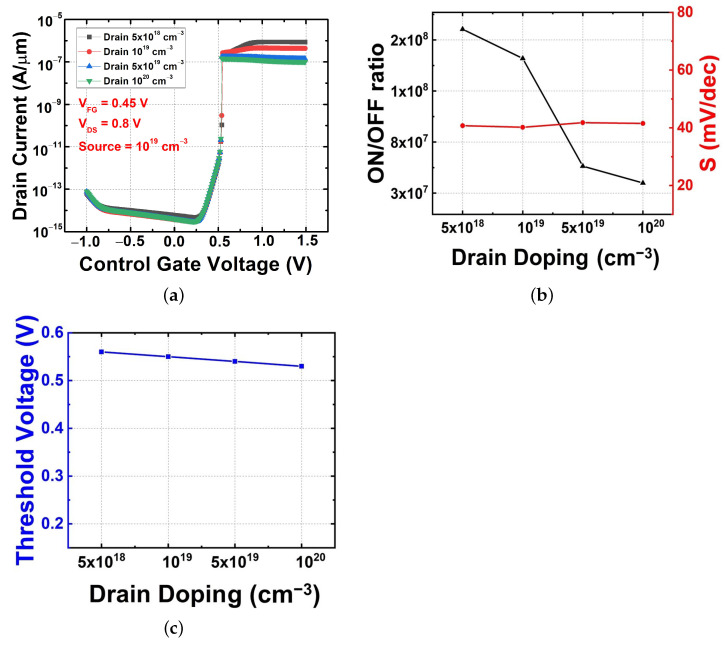
Electrical characteristics with drain doping variations. (**a**) Transfer characteristics, (**b**) on–off current ratio and subthreshold swing, and (**c**) threshold voltage.

**Figure 7 micromachines-13-00508-f007:**
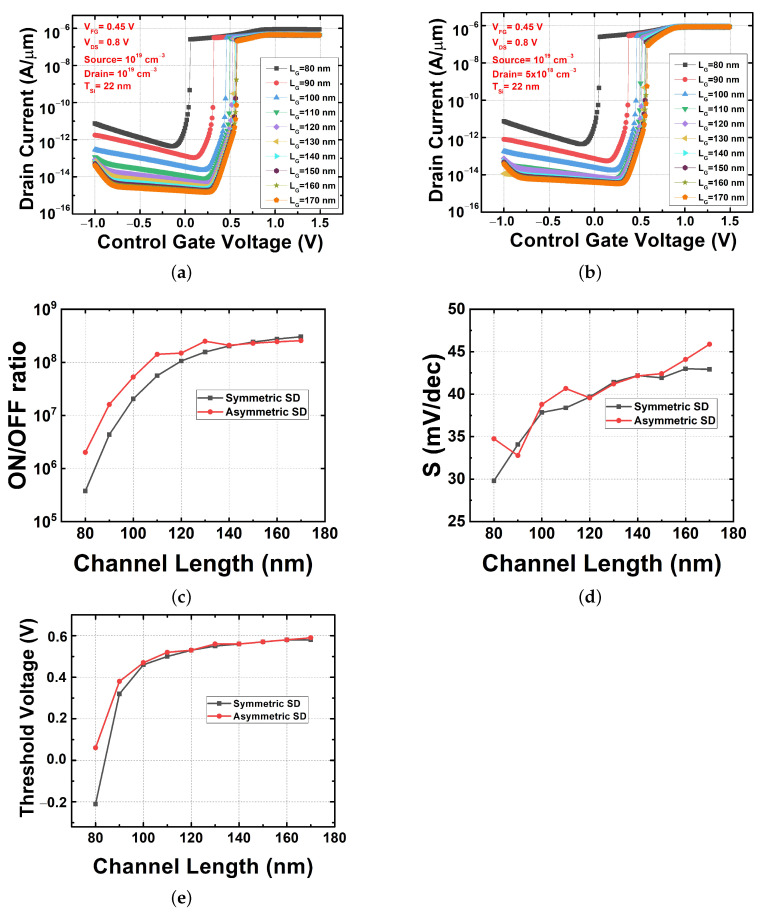
Electrical characteristics with channel length variation. (**a**) Transfer characteristics of symmetric source/drain FBFET, (**b**) transfer characteristics of asymmetric source/drain FBFET, (**c**) on–off current ratio in both FBFETs, (**d**) subthreshold swing in both FBFETs, and (**e**) threshold voltage in both FBFETs.

**Figure 8 micromachines-13-00508-f008:**
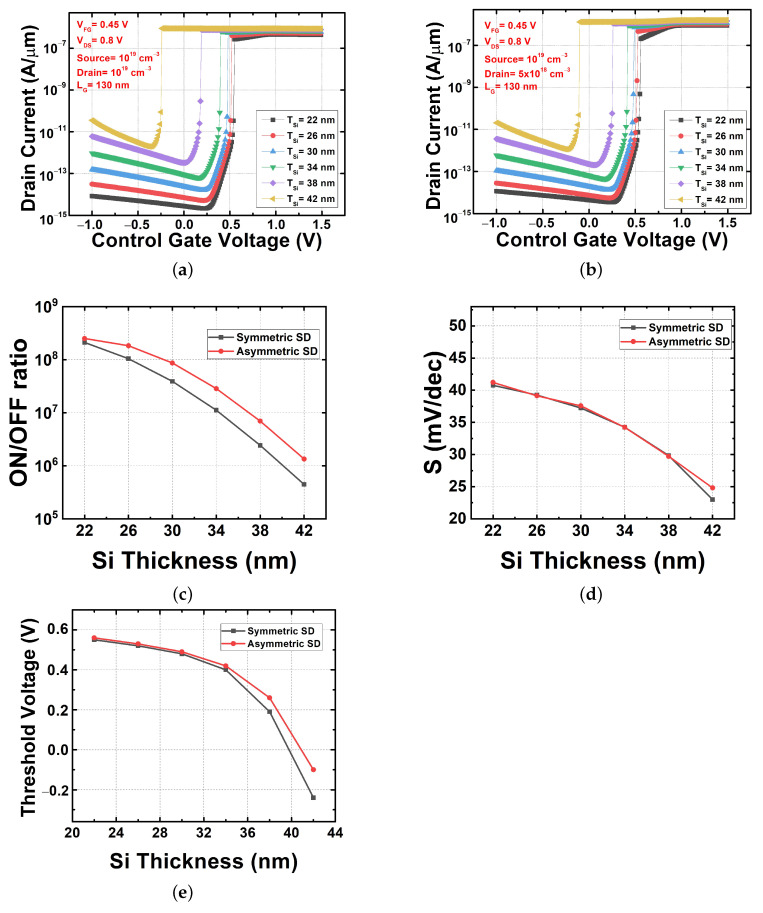
Electrical characteristics with channel thickness variation. (**a**) Transfer characteristics of symmetric source/drain FBFET, (**b**) transfer characteristics of asymmetric source/drain FBFET, (**c**) on–off current ratio in both FBFETs, (**d**) subthreshold swing in both FBFETs, and (**e**) threshold voltage in both FBFETs.

**Table 1 micromachines-13-00508-t001:** Default parameters of the FBFET simulation structure.

Parameter	Value [unit]
Channel length (L${}_{G}$)	130 [nm]
Silicon channel thickness (T${}_{Si}$)	22 [nm]
Poly-silicon gate thickness (T${}_{G}$)	130 [nm]
Control gate/fixed gate length (L${}_{CG}$/L${}_{FG}$)	63.5 [nm]
Gate dielectric thickness (T${}_{OX}$)	3 [nm]
Source/drain length (L${}_{S}$/L${}_{D}$)	10 [nm]
Source/drain doping concentration	1 × 10${}^{19}$ [cm${}^{-3}$]
Channel doping concentration	1 × 10${}^{15}$ [cm${}^{-3}$]
Control gate/fixed gate doping concentration	1 × 10${}^{20}$ [cm${}^{-3}$]

## Data Availability

Not applicable.

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
