# Peer review of "Optimization of Feedback FET with Asymmetric Source Drain Doping Profile"

_micromachines, 2022, doi:10.3390/mi13040508_

Round 1

Reviewer 1 Report

The paper is quite well presented. To complete the study, the authors are advised to add transient simulations so the switching behavior can be studied more easily.

Author Response

Thank you for the valuable comments. Here we send the answer sheet. Please see the attachment.

Thank you very much. 

Reviewer 2 Report

The manuscript conducted a TCAD simulation to optimize the design of asymmetric doping FBFET.  The device performance of on-off current ratio, VTH, and S was discussed with varied doping and dimensional conditions. It's a systematic study, and a lot of details were compared, but I would like to see more/deeper discussion about the following details:

(1) How the mesh is set? 
(2) Would the device structure be practical to be fabricated?
(3) Is there any similar device structure of FBFET reported? If so, could the author list a table to compare the performance with the reported ones?
(4) What's the intrinsic physic reason for the high on-off current ratio? And what are the dominant factors to achieve high performance? Why doping and dimensional conditions were discussed, and what conclusions or guidances can be achieved for future device optimization?
(5) Although it's a 2D simulation, could the author show the 3D device structure? And how the device is connected with the outer power supply? What's the basic working principle of FBFET? Why it can be called FBFET?

Author Response

(The authors gave the same response as above.)

Round 2

Reviewer 1 Report

Dears authors,

the addition of Ids vs time is welcome and would allow the reader to assess the transient behavior of the device regarding. If time permits, overlapping the control voltage and the output current on the same plot upon device switching would be appreciated so capacitances can be assessed.

The addition of the mesh structure is also welcome but is difficult to read. It also seems that the meshing is uniform in the whole structure. It would be best to indicate the minimum and maximum mesh sizes. 

Author Response

Thank you for the comment. Here we submit the answer sheet. 

Thank you. 

This manuscript is a resubmission of an earlier submission. The following is a list of the peer review reports and author responses from that submission.

Round 1

Reviewer 1 Report

The manuscript proposes a simulations study of an alternative FET architecture.  I do not recommend it for publication for the following reasons.

1) The quality of the writing is very poor. It is not only a matter of grammar, but, rather, the whole organization of most sentences is wrong. I could understand at best a sentence among two. Manuscripts of this quality should not be submitted. After the first page I was about to give up with the review. It is not even clear to me whether or not this device is a tunnel-FET type of device (I suspect it is).

2) Being a work based on modeling, description of the models and model parameters is mandatory. For example, which model is used for BTBT? Using which parameters? What is the impact of these parameters on the results? Same for SRH: how much the choice of the SRH times influences Ioff? What are the “quantum insight models” employed?

3) I cannot map Fig.1a with the terms used in the following: what is the mapping between G1 and G2 with CG and FG?

4) The choice of the geometry is not consistent with the aim of replacing nanoscale CMOS: here the channel length is 130nm, the silicon thickness 22nm and the EOT=3nm. Also the requested bias (0.8V) is quite high, since CMOS is heading toward sub 0.5V supply. Even as such high biases, Ion is quite low.

5) Vds is optimized at the beginning for a given geometry and doping, but it is not obvious that the same value is optimum when the device is changed

Reviewer 2 Report

The authors analyze the Feedback FET with different construction parameters. Paper presents the optimization of the device. The topic is interesting and currently under study by different researchers. It needs to be deeply investigated and have proper attention. Although interesting, the paper is poorly introduced and needs to be improved.

General suggestions:

- extensive English editing of language and style is strongly suggested,

- introduce clearly in the introduction the aim of the study, what you want to communicate in this paper, and why it is important to the community,

- add some general information about the operation principle of the Feedback FET,

- some information about the proposed structure is missing (how LG1 and LG2 are related to the parameters in table 1? which one is the control gate?)

- in Fig. 6 and 7, legend needs to be fixed (source and drain doping parameters are the same for symmetric and asymmetric configuration),

- describe clearly how you calculate the Savg, Vth and ON/OFF current ratio.

In general, the presented manuscript is a short collection of simulation results and extracted parameters. Some more insightful analysis of the device physics is needed.

Additionally, add some band diagram plots in the device cross-section to explain the device behavior and strengthen the analysis.

The aim of this journal is to encourage scientists to publish their theoretical and experimental results in as much detail as possible. There is no restriction on the length of the paper. Please, take this into consideration and do not downgrade the results of your work.